# SSEA3 and Sialyl Lewis a Glycan Expression Is Controlled by B3GALT5 LTR through Lamin A-NFYA and SIRT1-STAT3 Signaling in Human ES Cells

**DOI:** 10.3390/cells9010177

**Published:** 2020-01-10

**Authors:** Bi-He Cai, Hsueh-Yi Lee, Chi-Kan Chou, Po-Han Wu, Hsiang-Chi Huang, Chia-Chun Chao, Hsiao-Yu Chung, Reiji Kannagi

**Affiliations:** 1Institute of Biomedical Sciences, Academia Sinica, Taipei 11529, Taiwan; bigbiha@ibms.sinica.edu.tw (B.-H.C.); sakon@ibms.sinica.edu.tw (H.-Y.L.); bio909@gmail.com (C.-K.C.); hank726jr@gmail.com (P.-H.W.); hsiangchi2009@gmail.com (H.-C.H.); smoldwolf@gmail.com (C.-C.C.); shirleychung1987@gmail.com (H.-Y.C.); 2Department of Biology and Anatomy, National Defense Medical Center, Taipei 11490, Taiwan; 3Department of Medicine, College of Medicine, I-Shou University, Kaohsiung 82445, Taiwan; 4Genomics Research Center, Academia Sinica, Taipei 11529, Taiwan

**Keywords:** SSEA3, sialyl Lewis a, embryonic stem cell, differentiation, NFYA, STAT3

## Abstract

*B3GALT5* is involved in the synthesis of embryonic stem (ES) cell marker glycan, stage-specific embryonic antigen-3 (SSEA3). This gene has three native promoters and an integrated retroviral long terminal repeat (LTR) promoter. We found that *B3GALT5*-LTR is expressed at high levels in human ES cells. *B3GALT5*-LTR is also involved in the synthesis of the cancer-associated glycan, sialyl Lewis a. Sialyl Lewis a is expressed in ES cells and its expression decreases upon differentiation. Retinoic acid induced differentiation of ES cells, decreased the short form of NFYA (NFYAs), increased phosphorylation of STAT3, and decreased B3GALT5-LTR expression. NFYAs activated, and constitutively-active STAT3 (STAT3C) repressed B3GALT5-LTR promoter. The NFYAs and STAT3C effects were eliminated when their binding sites were deleted. Retinoic acid decreased the binding of NFYA to B3GALT5-LTR promoter and increased phospho-STAT3 binding. Lamin A repressed NFYAs and SSEA3 expression. SSEA3 repression mediated by a SIRT1 inhibitor was reversed by a STAT3 inhibitor. Repression of SSEA3 and sialyl Lewis a synthesis mediated by retinoic acid was partially reversed by lamin A short interfering RNA (siRNA) and a STAT3 inhibitor. In conclusion, *B3GALT5*-LTR is regulated by lamin A-NFYA and SIRT1-STAT3 signaling that regulates SSEA3 and sialyl Lewis a synthesis in ES cells, and sialyl Lewis a is also a ES cell marker.

## 1. Introduction

The gene *B3GALT5* encodes β-1,3-galactosyltransferase 5, which is involved in the synthesis of the globoseries glycolipids such as the human embryonic stem (ES) cell markers SSEA3 and SSEA4 and the tumor marker Globo-H (Appendix A) [1,2,3,4]. In addition, B3GALT5 is involved in the synthesis of the type-1-chain lactosamine glycans, which include Lewis a, Lewis b, sialyl Lewis a, and SSEA5 [5] (Appendix A). Sialyl Lewis a is a tumor marker for cancers of the digestive organs such as the colon [6], and SSEA5 is an ES cell marker [7]. SSEA-3 and SSEA4 are essential for cancer cell survival and metastasis through association with FAK and CAV1 to induce AKT signaling and to inhibit Fas-dependent cell death [8]. Sialyl Lewis a is essential for cancer cell migration and invasion through selectin-mediated signaling [6]. Sialyl Lewis a also modifies fibulin-3 to enhance EGFR signaling for activation of the PI3K/Akt/mTOR pathway for cell growth and proliferation [9]. Therefore, B3GALT5 is the key enzyme producing these cancer-related glycans such as SSEA-3 and sialyl Lewis a.

The *B3GLAT5* gene has three native promoters and one long terminal repeat (LTR) promoter [10,11]. An endogenous retrovirus is thought to have integrated its LTR promoter and an exon (exon 1) into the *B3GLAT5* gene. *B3GLAT5*-LTR is found only in the genomes of apes, old world monkeys, and humans [12] and is rarely expressed in normal human tissues, except in the colon [11]. SSEA3 is a marker related to the depletion of non-reprogramming cells, which are derived from induced pluripotent stem cells [13], and SSEA3 and SSEA5 are also involved in the removal of undifferentiated or incompletely differentiated cells prior to transplantation to avoid teratoma formation [5,14].

Sialyl Lewis a, Lewis a, and SSEA5 have also been identified in human H9 ES cells, and synthesis of these glycans is reduced by retinoic acid (RA) treatment [7]. Conversely, Lewis b has not been detected in undifferentiated or RA-differentiated cells [7]. B3GALT5, fucosyltransferase 2 (FUT2), and FUT3 are the key enzymes involved in the synthesis of sialyl Lewis a, Lewis a, SSEA5, and Lewis b (Appendix A), with the FUT2 gene having the functional dominant allele Se (secretor) and the non-functional recessive allele, se, while FUT3 has the functional dominant allele Le (Lewis) and the non-functional recessive allele, le [15]. The Lewis blood-group system comprises the three phenotypes Le(a+b-), Le(a-b+), and Le(a-b-). When the activity of FUT2 is substantial, the substrate Lc4 glycan is mostly converted into the type-1 chain H-antigen or Lewis b glycan instead of the Lewis a or sialyl Lewis a glycan [16]. The Le(a+b+) phenotype is transiently observed in infants [17], and some adults have the weak allele Se [18]. Humans with the Le(a+b-) phenotype have more sialyl Lewis a than do those with the Le(a-b+) phenotype [19]. Others have the Le(a-b-) phenotype, including 3.2% of the Chinese population [20], for which no sialyl Lewis a, Lewis a, or Lewis b is found in any organ [21].

Clarification of the regulatory mechanism of *B3GALT5*-LTR expression would contribute to our understanding of how the aforementioned human ES cell markers are synthesized and provide a clearer view of how enrichment of induced pluripotent stem cells or the decrease in the teratoma-forming rate occurs, which would be of use in the improvement of regenerative medical procedures. In this study, we found that human ES cells express *B3GALT5*-LTR and that RA-mediated differentiation signaling reduces the level of *B3GALT5*-LTR transcript to a greater extent than the native B3GALT5 transcripts. We also identified key transcription factors and upstream signals that regulate the *B3GALT5*-LTR promoter and SSEA3 expression in human ES and embryonal carcinoma (EC) cells during differentiation. Finally, using an antibody to stain silyl Lewis a in human ES and EC cells, we also found that, in addition to SSEA3, sialyl Lewis a acts as a human ES cell marker and is influenced by the same signals that modulate *B3GALT5*-LTR expression.

## 2. Materials and Methods

### 2.1. Cell Lines, Cell Culture, and Drug Treatment

An ES cell line of Taiwanese origin (TW3) [22] was obtained from the Bioresource Collection and Research Center (BCRC) and cultured on MEF feeder cells with DMEM/F12 containing 15% (*v*/*v*) KO serum replacement (Thermo Fisher, Carlsbad, CA, USA), 0.1 mM 2-mercaptoethanol, 0.1 mM non-essential amino acids, 4 ng/mL bFGF, and 1 mM l-glutamine. The H1 octamer-binding transcription factor 4-enhanced green fluorescent protein (Oct4-EGFP) ES cell line [23] was obtained from WiCell and cultured on the Corning Matrigel matrix (human ES cell-qualified) with StemFlex medium (Thermo Fisher). Oct4-EGFP ES cells were cultured with G418 3000 μg/mL (Millipore, Billerica, MA, USA) to maintain them in their undifferentiated state. NCCIT, DLD1, and SKW6.4 cells were maintained in RPMI 1640 medium (Thermo Fisher) supplemented with 10% fetal bovine serum (Thermo Fisher), 100 U/mL penicillin, and 100 μg/mL streptomycin. NT2, 2102Ep, and HT29 cells were maintained in DMEM medium (Thermo Fisher) supplemented with 10% fetal bovine serum (Thermo Fisher), 100 U/mL penicillin, and 100 μg/mL streptomycin. All cells were cultured at 37 °C under a humidified 5% CO_2_ atmosphere. For ES and EC cell differentiation, the cells were treated with 10 μM RA (Sigma-Aldrich, St. Louis, MO, USA) in DMSO for 7 days with pure DMSO serving as the control. G418 was removed from the H1 Oct4-EGFP cells during RA-induced differentiation. Where indicated, H1 Oct4-EGFP cells were treated with 10 nM of the SIRT1 inhibitor EX-527 (Sigma-Aldrich), 100 μM of the STAT3 inhibitor S3I-201 (Santa Cruz Biotechnology, Dallas, Texas, USA), 5 μM of the STAT3 inhibitor Stattic (Santa Cruz Biotechnology, Dallas, Texas, USA) or 5 nM bortezomib (Toronto Research Chemicals, Toronto, ON, Canada) for 7 days (G418 was included in each case).

### 2.2. Flow Cytometry

Cultured cells were harvested (5 × 10^5^ cells per tube), centrifuged at 300× *g* for 5 min, and incubated with an anti-SSEA3 (Rat IgM, R&D Systems, Minneapolis, MN, USA) or anti-sialyl Lewis a (CA19-9 [116-NS-19-9] Mouse IgG1, Thermo Fisher) at 4 °C for 30 min. Then, cells were washed with 1 mL of buffer for fluorescence-activated cell sorting (FACS; phosphate-buffered saline containing 2% fetal bovine serum (Thermo Fisher)) and stained with Alexa Fluor 647-conjugated goat anti-rat IgM secondary antibody (Thermo Fisher), FITC-conjugated goat anti-rat IgM secondary antibody (Jackson ImmunoResearch, West Grove, PA, USA), or APC-conjugated goat anti-mouse IgG secondary antibody (BioLegend, San Diego, CA, USA) at 4 °C for 30 min. Next, the cells were washed twice with 1 mL FACS buffer, resuspended in 0.4 mL of the buffer, and kept in the dark on ice until FACS analysis (the cells were first passed through a mesh and then subjected to flow cytometry, Attune NxT, Thermo Fisher).

### 2.3. Plasmid Construction

Full-length coding sequences for the short form of NFYA (NFYAs; NCBI accession number NM_021705.3) and the STAT3 gene (NCBI accession number NM_139276.2) were amplified from NT2 cDNA with the use of the following primer sets: 5′-ATGGAGCAGTATACAGCAAACAG-3′ and 5′-TTAGGACACTCGGATGATCTGT (NFYAs), and 5′-ATGGCCCAATGGAATCAGCTACA-3′ and 5′-TCACATGGGGGAGGTAGCGC-3′ (STAT3). The PCR products were then cloned into pcDNA3.0 (Invitrogen, Carlsbad, CA, USA). The NFYAm29 and STAT3C point-mutation constructs were created by site-directed mutagenesis (Phusion Site-Directed Mutagenesis kit, Finnzymes). The following primers were used: 5′-CAGCCTTCCGTGCCATGGC-3′ and 5′-CTGCAGGTGGACGATTTTTCTCTC-3′ (NFYAm29), and 5′-ACTGGTCTATCTCTATCCTGACATTCCCA-3′ and 5′-GGAGACACCAGGATACAGGTACAATCCATGATC-3′ (STAT3C). The PCR product of the full-length human B3GALT5-LTR promoter, which resides on chromosome 21 (GRCh38.p12 Primary Assembly. NC_000021.9: 39657153–39657326; from nucleotides −174 to −1) was amplified using NT2 genomic DNA as the template and the forward primer 5′-GGAGCCTGCAGCAGGCAGAGGC-3′ and reverse primer 5′-CGGGTCCAAAGGCCAGAGAGC-3′. The PCR products were purified by the EasyPrep Gel & PCR Extraction Kit (TOOLS, Taiwan). The purified PCR product was cloned upstream of the firefly luciferase reporter gene (Luc) in the pGL3-enhancer vector (Promega). The B3GALT5-LTR HNF-1d, -NFYd, and -STAT3d constructs were created by site-directed mutagenesis. The following primers were used: 5′-CAGCCAAGTTGACACCTAAAAGTAACC-3′ and 5′-TAGAGAACTGGTAAAGCATTATTTCTGGG-3′ (B3GALT5-LTR HNF-1d), 5′-CTCTGGAAACACCTTCACAAACACA-3′ and 5′-TCAGTGGGCTGAGTG GGGAG-3′ (B3GALT5-LTR NFYd), and 5′-ACCTTCACAAACACACCCAGAAATAATG-3′ and 5′- AGATTGGCTGTGAGTCAGTGGGC-3′ (B3GALT5-LTR STAT3d). A tandem repeat NFY response construct containing two repeats of TAACCAATCA sequences was cloned into the SmaI site of the pGL3 promoter as previously described [24]. pcDNA3.1-NFYAl and lamin A clones were obtained from Genscript and Sinobiological, respectively. The sequences of all constructs were verified by DNA sequencing.

### 2.4. Transfection of Cells with Plasmids or Short Interfering RNAs (siRNAs)

For liposome-mediated transfection of cultured cells (5 × 10^5^) with plasmids, the cells were plated into a well of a 6-well dish one day before transfection. The following day, 2 µg of a plasmid was mixed with 200 µL Opt-MEM medium (Thermo Fisher); then, 4 µL of X-tremeGENE HP DNA Transfection reagent (Roche) was added, with incubation at room temperature for 20 min. Each mixture was then added into the cells. For electroporation-mediated transfection of cultured cells (1 × 10^6^) with each plasmid, cells were added into 90 µL BTXpress electroporation buffer (BTX, Holliston, MA) that contained 3 µg of a plasmid. The mixture was transferred into a 2-mm BTX Gap cuvette and with subsequent electroporation (140 V, 750 Ω, 1100 μF capacitance, 13 ms, one pulse) using the BTX Gemini X2 Electroporation system. After electroporation, the mixture was collected from the cuvette and added into the wells. For transfection of cultured cells (5 × 10^5^) with a siRNA, the cells were plated into wells of a 6-well dish one day before transfection. The next day, 2 μL of control siRNA (50 μM; siGENOME Non-Targeting siRNA, Dharmacon) or lamin A-specific siRNA (siGENOME lamin A/C, Dharmacon) was mixed with 200 µL Opt-MEM medium, after which 14 µL of Lullaby Stem siRNA Transfection reagent (OZ Biosciences) was added into each mixture; mixtures were incubated at room temperature for 20 min. Finally, each mixture was individually added into a cell preparation.

### 2.5. Luciferase Assay

Cells were co-transfected with the B3GALT5-LTR firefly luciferase plasmid or the tandem repeat NFY response element RE on the pGL3 promoter vector and the pRL-SV40 Renilla luciferase plasmid, respectively (Promega). Cells were harvested 48 h post-transfection into 0.25 mL of the reporter lysis buffer and assayed for gene expression with the Dual-Luciferase Reporter Assay system (Promega, Madison, WI, USA). Firefly luciferase activity was normalized to Renilla luciferase activity, and the data are presented as the mean ± standard deviation of three independent experiments, each of which was performed in triplicate.

### 2.6. Western Blotting

Total cellular protein extract (20 μg) was subjected to SDS-PAGE (10% acrylamide). The separated proteins were transferred onto a polyvinylidene difluoride membrane, which was then blocked with 5% skim milk in phosphate-buffered saline and [0.05% (*v*/*v*)] Tween 20. Membranes were probed with antibodies against NFYA (sc-10779 rabbit polyclonal; Santa Cruz Biotechnology), STAT3 (sc-482 rabbit polyclonal; Santa Cruz Biotechnology), phospho-STAT3 (STAT3-p; phosphorylated Tyr705; #9145 rabbit monoclonal; Cell Signaling Technology), SIRT1 (sc-15404 rabbit polyclonal; Santa Cruz Biotechnology), lamin A/C (10298-1-AP rabbit polyclonal, Proteintech), β-tubulin (10094-1-AP rabbit polyclonal, Proteintech, Chicago, IL, USA) or GAPDH (10494-1-AP rabbit polyclonal, Proteintech). Goat anti-rabbit IgG conjugated with horseradish peroxidase (Santa Cruz Biotechnology) served as the secondary antibody. Chemiluminescent signals were detected using reagents from an Amersham ECL Western Blotting Detection kit.

### 2.7. ChIP Assay

Chromatin immunoprecipitation (ChIP) was performed according to the manufacturer’s instructions (EZ-Magna ChIP A/G Chromatin Immunoprecipitation kit; Millipore) using anti-NFYA (sc-10779; Santa Cruz Biotechnology) and anti-STAT3 (sc-482; Santa Cruz Biotechnology) as shown in Figure 3C and anti-NFYA (sc-17753; Santa Cruz Biotechnology) and anti-STAT3 (#9139; Cell Signaling Technology) (see Figure 3D). Fixed DNA was sheared with a Bioruptor Pico (Diagenode, Liège, Belgium), and precipitated DNA was quantified with a Bio-Rad CFX96 Real-Time Thermal Cycler.

### 2.8. Real-Time Quantitative PCR and Primers

Total cellular RNA was extracted with TRIzol reagent (Invitrogen). First-strand cDNA from mRNA was prepared from 1 μg total RNA with reagents of the Maxima H Minus FirstStrand cDNA Synthesis kit (Thermo Scientific, Waltham, MA, USA). For conventional qPCR, cDNA samples or DNA samples from ChIP assays were mixed with EvaGreen Supermix (Bio-Rad, Hercules, CA, USA) and primers (Appendix A). Amplification of each qPCR product was monitored using a CFX Connect Real-Time PCR System (Bio-Rad). Relative transcript levels were calculated as 2^−ΔΔCT^. The percentage of the B3GALT5-LTR transcript was calculated as (2^−(B3GALT5-LTR CT) – (B3GALT5 total CT)^) × 100%.

### 2.9. Statistical Analysis

Potential statistical differences between two groups were assessed with the Student’s *t*-test (two tailed). All results are presented as the mean ± SD. A *p*-value of less than 0.05 was considered statistically significant (*, *p* < 0.05; **, *p* < 0.01; ***, *p* < 0.001; and ns: not statistically significant).

## 3. Results

### 3.1. B3GALT5-LTR is Highly Expressed in ES and EC Cells

Expression of *B3GALT5*-LTR in colon tissue contributes up to 74% of the total *B3GALT5* expression [11], and we found that *B3GALT5*-LTR accounted for ~80% of the total expressed *B3GALT5* in the colon cancer cell line HT29 (Figure 1A). We also found that the *B3GALT5*-LTR gene was highly expressed in EC (NT2) and ES (TW3) cells at levels comparable to those of the colon cancer cell lines DLD1 and HT29, whereas *B3GALT5*-LTR was not expressed in B cells (SKW6.4; Figure 1B). Expression of the *B3GALT5*-LTR gene in NT2 cells accounted for ~80% of total *B3GALT5* expression, and its expression in TW3 cells was ~40% (Figure 1A). Compared to *B3GALT5*-LTR, expression of other transcripts through three native promoters was relatively low in both H1 and 2102Ep cells (Figure 1C). We treated the ES and EC cells with RA to induce differentiation (Figure 1D), after which the amount of *B3GALT5*-LTR expressed decreased to ~15% of the total expressed B3GALT5 (Figure 1E). The glycan SSEA3, which requires *B3GALT3* for its synthesis, serves as a stem-cell marker [1]. Inhibition of SSEA3 and SSEA4 synthesis by the inhibitors of sphingolipid and glycosphingolipid synthesis, D-PDMP or ISP-1, does not influence Oct4 expression [25], and the ES cell line, H1 Oct4-EGFP, has been used for evaluating ES-cell differentiation [23]. We found that the amount of SSEA3 and GFP was suppressed by RA treatment of H1 Oct4-EGFP cells (Figure 1F).

### 3.2. NFYA and STAT3, but Not HNF-1, Are the Transcription Factors that Regulate B3GALT5-LTR Expression in ES and EC Cells

To identify the transcription factors that might induce LTR promoter–controlled expression of *B3GALT5* in EC and ES cells, we cloned the LTR promoter region of *B3GALT5*-LTR and conjugated it to the Luc reporter gene after individually deleting the possible transcription factor–binding sites, HNF-1 and NFY, that might be involved in *B3GALT5*-LTR expression (Figure 2A). We found that deletion of the NFY, but not the HNF-1-binding site in the LTR promoter reduced the reporter activity of the construct in NT2 cells, whereas deletion of NFY or HNF-1 nearly eliminated the reporter activity in HT29 cells (Figure 2B). HNF-1 maintains the high level of expression of B3GALT5-LTR in colon cancer cells [26]. In EC and ES cells, however, HNF-1 was hardly expressed (Appendix A). The rat albumin promoter contains a NFY-binding site at nucleotide −80 and a HNF1-binding site at −60 [27]. During development, NFY binds to and activates the rat albumin promoter when the concentration of HNF1 is limited [27], indicating that NFY can replace the function of HNF1 in activating promoters. NFY contains three isoforms, namely, NFYA, NFYB, and NFYC, of which NFYA has a short form (NFYAs) and a long (NFYAl) form [28]. NF-Y heterotrimer is composed of a histone fold domain dimer NF-YB/NF-YC and NF-YA, which contains a DNA specific binding domain [29]. The amount of NFYAs, but not of NFYAl, is reduced upon differentiation of mouse ES cells [30]. We found that NFYAs is the major isoform of the NFYA subunit in NFY in human ES and EC cells (Appendix A) and is reduced after differentiation of ES or EC cells (Appendix A).

In addition to the NFY-binding site, a STAT3-binding site was predicted in the *B3GALT5*-LTR promoter (Figure 3A). To elucidate which signaling cascade(s) controls *B3GALT5*-LTR expression during differentiation, we investigated the post-translational phosphorylation of STAT3, which acts upstream of *B3GALT5*-LTR gene expression. First, we treated NT2 cells with RA or pure DMSO (control) and then stained them with antibodies against NFYA, total STAT3, and STAT3-p. We found that the amount of NFYAs, but not the amount of NFYAl, was reduced by RA treatment whereas the amount of STAT3-p was increased (Figure 3B). RA causes NT2 cells to differentiate into astrocytes [31], and differentiation is accompanied by increased phosphorylation at Y705 of STAT3 [32]. We found that phosphorylation of STAT3 was enhanced by RA treatment in NT2, NCCIT, and H1 Oct4-EGFP cells (Figure 3B). Phosphorylation at Y705 results in STAT3 dimerization and nuclear translocation as the activated form [33]. The activity of a modified form of STAT3, STAT3C, mimics that of STAT3-p [34]. In STAT3C, substitution of two cysteines, A661C and N663C, in the C-terminal loop of the Src homology 2 domain produces a molecule that dimerizes spontaneously. Because the transfection efficiency of NT2 cells is poor compared to that of 2102Ep cells (Appendix A), we used the 2102Ep cells in a STAT3C-overexpression experiment. 2102Ep cells do not differentiate when treated with RA [35], but this cell line expresses human ES cell markers [36], making it a useful model for studying the effect of various drugs on their expression. Overexpression of STAT3C repressed *B3GALT5*-LTR expression in 2102Ep cells, and repression of *B3GALT5*-LTR by STAT3C was diminished after deletion of the STAT3-binding site on the *B3GALT5*-LTR promoter (Appendix A). As shown by ChIP assays, NFYA bound to the *B3GALT5*-LTR promoter in undifferentiated NT2 cells, and STAT3 bound to the *B3GALT5*-LTR promoter in differentiated NT2 cells (Figure 3C). In addition, NFYA bound to the *B3GALT5*-LTR promoter more robustly in undifferentiated H1 cells than differentiated cells, and STAT3 bound to the *B3GALT5*-LTR promoter more robustly in differentiated H1 cells than undifferentiated cells (Figure 3D). NFYAm29, a dominant-negative mutant of NFYA, can form the heterotrimer with NFYB and NFYC but cannot bind DNA [37]. Electroporation of 2102Ep cells with a plasmid containing either the NFYAm29 or the STAT3C construct resulted in repression of SSEA3 synthesis (Figure 4A–C).

### 3.3. RA-Mediated Lamin A-NFYA Pathway Regulates B3GALT5-LTR Promoter and Represses Production of SSEA3 and Sialyl Lewis a

NFYAs and NFYAl have different transcriptional activities [41], as they activate or repress different genes during muscle cell differentiation [42]. Overexpression of NFYAs enhanced SSEA3 expression and B3GALT5-LTR promoter activity, while overexpression of NFYAl repressed SSEA3 and B3GALT5-LTR promoter activity (Figure 5A–C). Activation by NFYAs or repression by NFYAl was diminished after deletion of the NFY-binding site on the *B3GALT5*-LTR promoter (Figure 5D). The amount of NFYA and its activity is decreased by lamin A [43,44], and lamin A is an ES cell differentiation marker [45]. Lamin A was downregulated by RA treatment of NCCIT and H1 Oct4-EGFP cells (Appendix A). Overexpression of lamin A repressed the activities of the NFYA responsive element and the *B3GALT5*-LTR promoter (Figure 5E). The expression of NFYAs and SSEA3 was also repressed by lamin A (Figure 5A,B). RA-mediated repression of SSEA3 and sialyl Lewis a expression were partially reversed by lamin A siRNA (Figure 5F,G, and Appendix A showing KD efficiency).

### 3.4. RA-Mediated SIRT1-STAT3 Pathway Regulates B3GALT5-LTR Promoter Activity and Thereby Represses SSEA3 and Sialyl Lewis a Synthesis

SIRT1 represses STAT3 activity [46], and SIRT1 is highly expressed in ES cells with its expression being reduced after differentiation [47]. SIRT1 was downregulated by RA treatment of NT2 and H1 Oct4-EGFP cells (Appendix A). The SIRT1 inhibitor, EX-527, repressed SSEA3 and sialyl Lewis a synthesis and *B3GALT5*-LTR promoter activity, while the STAT3 inhibitor, S3I-201, enhanced SSEA3 and sialyl Lewis a synthesis and *B3GALT5*-LTR promoter activity (Figure 6A–E). These actions of EX-527 were reversed by S3I-201 (Figure 6D,E). Bortezomib represses SIRT1 expression and increases the activity of STAT3 [48,49]. We also found that the amount of SSEA3, sialyl Lewis a, and *B3GALT5*-LTR promoter activity was reduced by bortezomib (Figure 6F–H), and the bortezomib-mediated repression of SSEA3, sialyl Lewis a, and *B3GALT5*-LTR promoter was reversed by Stattic, another STAT3 inhibitor. EX-527-mediated repression of SSEA3 and sialyl Lewis a was also partially reversed by Stattic (Appendix A). In addition, RA-mediated repression of SSEA3 was partially reversed by Stattic (Appendix A), and RA-mediated repression of sialyl Lewis a was partially reversed by S3I-201 (Appendix A).

Taken together, our results indicate that the *B3GALT5*-LTR gene is highly expressed in ES cells, and according to our findings, it appears that *B3GALT5*-LTR expression is under the control of the lamin A-NFYA and SIRT1-STAT3 pathways in human EC and ES cells (Figure 7).

## 4. Discussion

Expression of the *B3GLAT5* gene is controlled by three native promoters and one retroviral-inserted LTR promoter [10,11], and the protein product is identical regardless of which promoter is used [50]. The LTR promoter is the most active of these promoters in the colon [10], and HNF-1 is the key transcription factor that activates it in the colon [26]. In this study, we found that the *B3GALT5*-LTR transcript is also highly expressed in human ES and EC cells, yet almost no HNF-1 is produced in ES cells. Rather, we found that the transcription factors NFYA and STAT3 regulate *B3GALT5*-LTR expression in ES cells, and that upstream lamin A and SIRT1 signaling mediates NFYA and STAT3 activity, thereby modulating SSEA3 synthesis. RA is a cell-differentiation agent that induces ES cells to enter a neural lineage [51]. The level of SSEA3 was found to be reduced from 93.9% to 2.5% in human ES cells that had differentiated into neural progenitor cells, and from 93.9% to 62.9% in human ES cells that had differentiated into endodermal cells [52]. How differentiation signals induce human ES cells to become different mature cell lines via regulation of *B3GALT5*-LTR and/or native promoter activities, which then mediate changes in SSEA3 synthesis, needs further study.

SSEA3 and lamin A act as EC- and ES-cell undifferentiated and differentiated markers, respectively [1,3,45]. Lamin A does not co-stain with SSEA4 [45,53]. Expression of lamin A does not induce EC cells to differentiate or promote RA-mediated differentiation [54]. In addition, D-PDMP or ISP-1, when used to remove glucosylceramide, a precursor of the glycolipids SSEA3 and SSEA4, does not induce ES-cell differentiation or influence Oct4 expression [25]. In our undifferentiated H1 Oct4-EGFP cells maintained by G418, the amounts of SSEA3 and sialyl Lewis a were altered by drug treatment, although that of OCT4 was >90% (Figure 6A,D,E). These findings suggest that the stem-cell markers SSEA3, sialyl Lewis a, and lamin A are not mediators of differentiation. Lamin A interacts with NFYA to block NFYA activity [44]. Lamin A reduces NFYA expression when it binds to the NFY complex at the CCAAT box on the NFYA promoter [43]. However, when we overexpressed lamin A in 2102Ep cells, we found decreased expression of NFYAs and increased expression of NFYAl (Figure 5A). In benign endometrial tumors or low-grade G1 endometrial cancers, relatively large amounts of lamin A and NFYAl and smaller amounts of NFYAs have been observed, whereas the opposite state, i.e., high levels of NFYAs and low levels of lamin A and NFYAl, were found in high-grade endometrial cancers (G2-G3) [55]. These findings suggest that lamin A may also influence the levels of pre-mRNA splicing factors that can alter the NFYAs/NFYAl ratio. Several splicing factors including RBM47 [56], ESRP1/2 [57], CELF1 [58] and QKI5 [59] can influence the NFYAs/NFYAl ratio. However, how differentiation signaling, including that of lamin A, alters the expression of these splicing factors and causes a change in the NFYAs/NFYAl ratio needs further investigation (Figure 7).

STAT3 signal helps keep murine ES cells in an undifferentiated state [60,61] but does not appear to do so in human ES cells [62,63]. RA treatment is reported to induce STAT3 activation in human EC cells [32]. In this study, we also found that RA induced phosphorylation of the STAT3 Y705 in EC and ES cells (Figure 3B), suggesting that the differentiation signal is induced by RA phosphorylates and activates STAT3. STAT3 generally acts as a transcriptional activator, but it also acts as a transcription repressor through recruiting HDAC1 or HDAC8. We have now shown that, in the presence of STAT3C (the constitutively active STAT3), the LTR promoter–driven expression of *B3GALT5* is repressed (Appendix A). However, the composition of the STAT3-repression complex on the *B3GALT5*-LTR promoter needs further clarification.

The *B3GALT5*-LTR promoter is relatively short (174 bp) and does not contain a CpG island [26]. Recently, however, two CpG islands—denoted as site 1 and site 2—were identified upstream of the *B3GALT5*-LTR exon, and the methylation status of site 2 appears to influence *B3GALT5*-LTR expression in colon cancer cells [64]. The genome of the colon cancer cell line COLO-205 is highly methylated at site 2, and when the cells were treated with 5′AZA (inhibitor of DNA methylation), histone opening mark (histone 3 lysine acetylation) at the HNF-1 binding region of *B3GALT5*-LTR promoter is reduced [64]. The colon cancer cell line Huh-7 and pancreatic cancer cell lines express HNF-1 [64,65,66], although they do not express *B3GALT5*-LTR, which may be a consequence of no or minimal methylation on the aforementioned CpG islands at site 2 upstream of the *B3GALT5*-LTR exon [64]. These findings suggest that hypermethylation of site 2 may open the B3GALT5 LTR promoter region to loosen the histone to allow transcription factor binding. Examination of the methylation map containing the upstream B3GALT5-LTR exon (available at the UCSC Genome Browser database [67]) revealed that that sites 1 and 2 of the genomes of ES, EC, and colon cancer cells are highly methylated, although those of pancreatic cancer cells are minimally methylated or not methylated (Appendix A). This means that in addition to colon cancer cells, ES and EC cells should also have a looser compaction of histone on the *B3GALT5*-LTR promoter.

## 5. Conclusions

In this study, we demonstrated that B3GALT5-LTR is highly expressed in human ES cells, and its expression is not dependent on the activity of HNF-1, a key transcription factor needed for B3GALT5-LTR expression in colonic cells. In differentiated ES cells, NFYAs decreased, phospho-STAT3 increased, and *B3GALT5*-LTR expression decreased. NFYAs acts as activator, and STAT3 acts as a repressor to regulate *B3GALT5*-LTR promoter. Repression of SSEA3 and sialyl Lewis a synthesis mediated by retinoic acid was partially reversed by lamin A siRNA and STAT3 inhibitors. We discovered that sialyl Lewis a is an ES cell marker as well as SSEA3, and expression of these two glycans is under the control of B3GALT5-LTR through lamin A-NFYA and SIRT1-STAT3 signaling.

## Figures and Tables

**Figure 1 cells-09-00177-f001:**
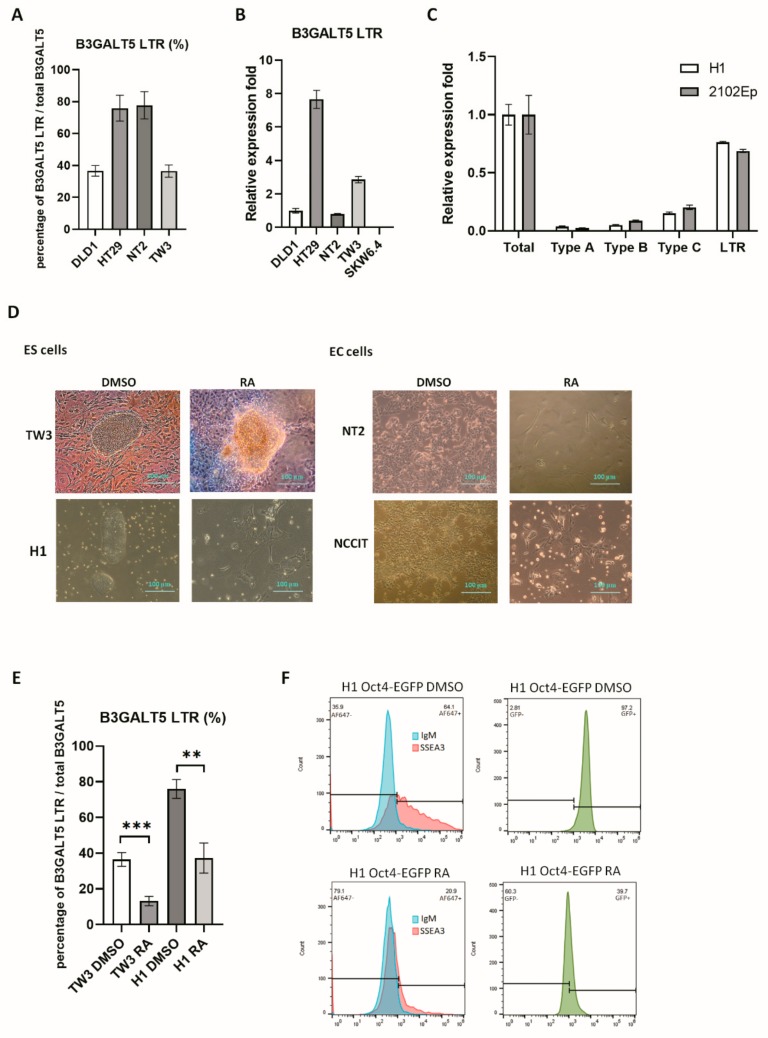
*B3GALT5*-long terminal repeat (LTR) expression in embryonic stem (ES) and embryonal carcinoma (EC) cell lines. (**A**) The expression of *B3GALT5*-LTR compared with that of total *B3GALT5* in colon cancer cell lines (DLD1, HT29), an EC cell line (NT2), and an ES cell line (TW3), reported as a percentage. (**B**) Normalized expression of *B3GALT5*-LTR in the cell lines is shown in Figure 1A and in a B-lymphoid cell line (SKW6.4). The DLD1 cell line amount was set to one. (**C**) Normalized expression of all *B3GALT5* transcripts in H1 and 2102Ep cells. Specific primers designed to amplify the *B3GALT5* coding region were used to quantify the total transcripts. Three native promoters with their own specific exon were used to design specific primers to quantify as Type A-C transcripts. *B3GALT5*-LTR transcript levels were higher than Type A-C transcripts. (**D**) Morphology of the undifferentiated and the retinoic acid (RA)-induced differentiated ES and EC cell lines. (**E**) Reduction of *B3GALT5*-LTR expression in ES cells (TW3 and H1) upon differentiation. (*p* < 0.05; **, *p* < 0.01; ***) (**F**) Octamer-binding transcription factor 4 (Oct-4)-controlled expression of enhanced green fluorescent protein (EGFP) was reduced after RA treatment for one week in H1 Oct4-EGFP cells. Stage-specific embryonic antigen-3 (SSEA3) was also decreased after RA treatment.

**Figure 2 cells-09-00177-f002:**
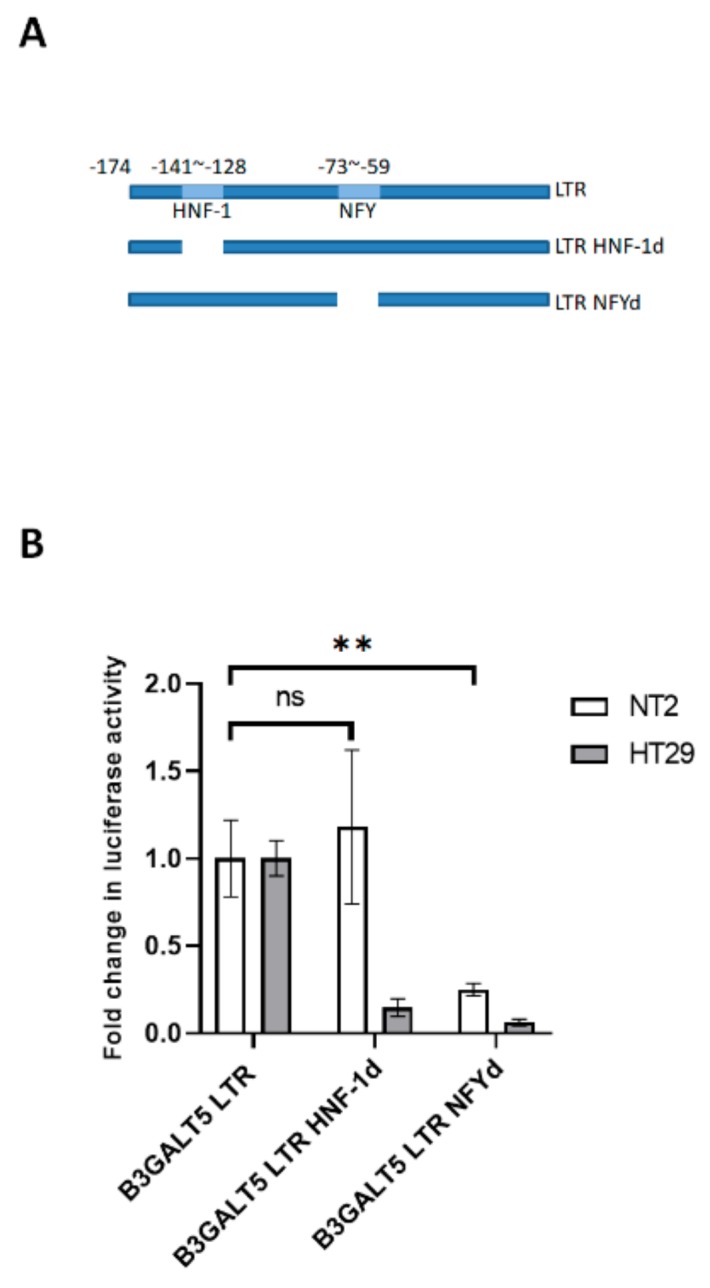
Transcription factors involved in B3GALT5-LTR expression. (**A**) The positions of the canonical HNF1- and NFY-binding sites on the *B3GALT5*-LTR promoter are shown. (**B**) Expression of the Luc-LTR reporter construct with a deleted HNF-1 binding site in its promoter (HNF-1d) did not change in NT2 cells but was reduced in HT29 cells. Luciferase activity was greatly reduced when NT2 and HT29 cells were transfected with Luc-LTR with a deleted NFY binding site (NFYd). (**, *p* < 0.01 and ns: not statistically significant).

**Figure 3 cells-09-00177-f003:**
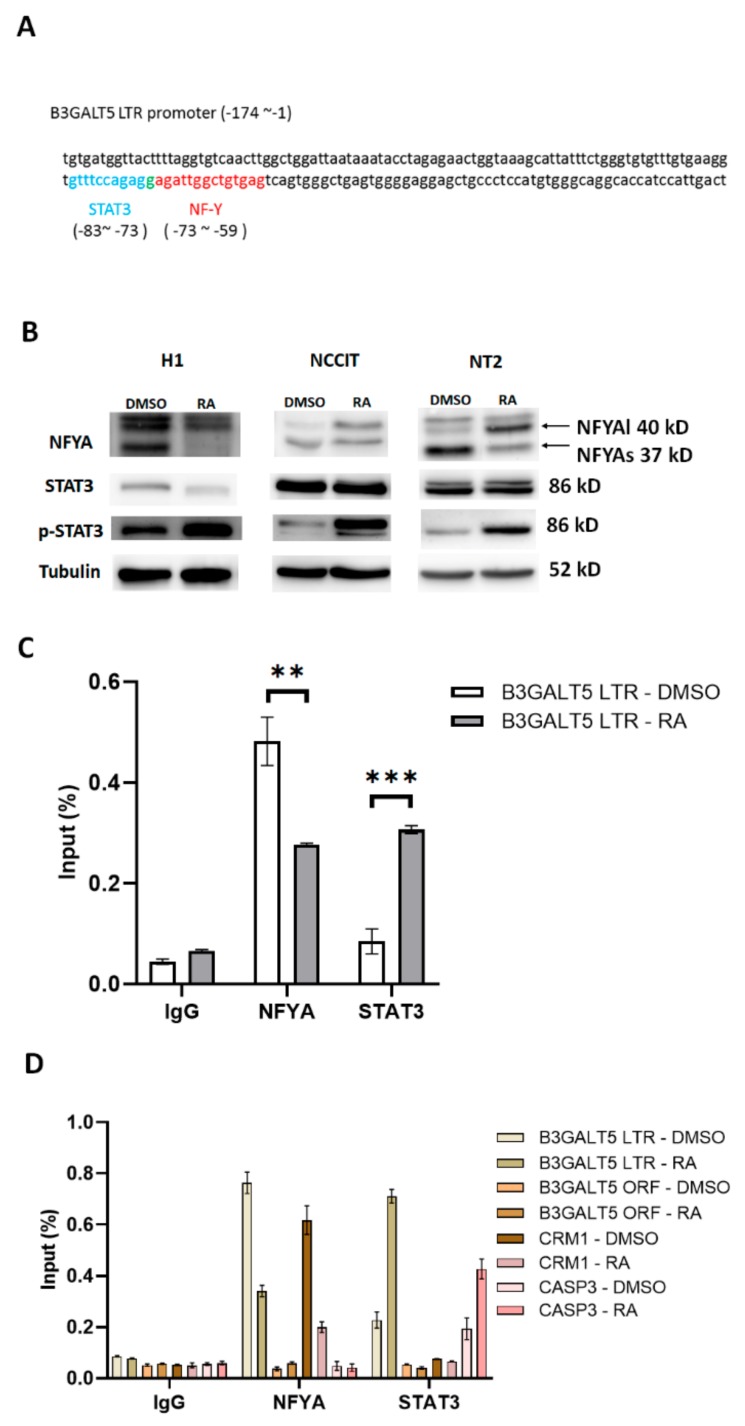
NFYA and STAT3 bind to the B3GALT5 promoter. (**A**) The positions of the canonical NFYA- and STAT3- binding sites in the *B3GALT5*-LTR promoter as predicted by the JASPAR database [38]. (**B**) Expression of NFYA isoforms and the STAT3 phosphorylation status in RA-treated ES and EC cells. Phosphorylation of STAT3 was markedly enhanced after RA treatment. Expression of NFYAs was reduced by RA treatment of ES and EC cells, and NFYAl expression was induced in EC cells but not in ES cells. Tubulin served as the loading control for these Western blots. (**C**) NFYA and STAT3 bound to the *B3GALT5*-LTR promoter as shown by chromatin immunoprecipitation (ChIP) assays in NT2 cells. NFYA bound to the *B3GALT5*-LTR promoter in undifferentiated NT2 cells (DMSO-treated cells) to a greater extent than in differentiated cells (RA-treated cells). STAT3 bound to the *B3GALT5*-LTR promoter in differentiated but not undifferentiated NT2 cells. One percent of the chromatin is labeled as input from each sample. (**, *p* < 0.01; ***, *p* < 0.001) (**D**) NFYA and STAT3 bound to the *B3GALT5*-LTR promoter as shown by ChIP assays in H1 cells. NFYA bound to the *B3GALT5*-LTR promoter in undifferentiated H1 cells (DMSO-treated cells) to a greater extent than in differentiated cells (RA-treated cells). STAT3 bound to the *B3GALT5*-LTR promoter in differentiated H1 cells to a greater extent than in undifferentiated cells. *CRM1* promoter contains a NFYA binding site [39] that is used as a positive control for NFYA binding, and *CASP3* promoter contains a STAT3 binding site [40] that is used as a positive control for STAT3 binding. *B3GALT5* open reading frame is used as a negative control for NFYA and STAT3 binding.

**Figure 4 cells-09-00177-f004:**
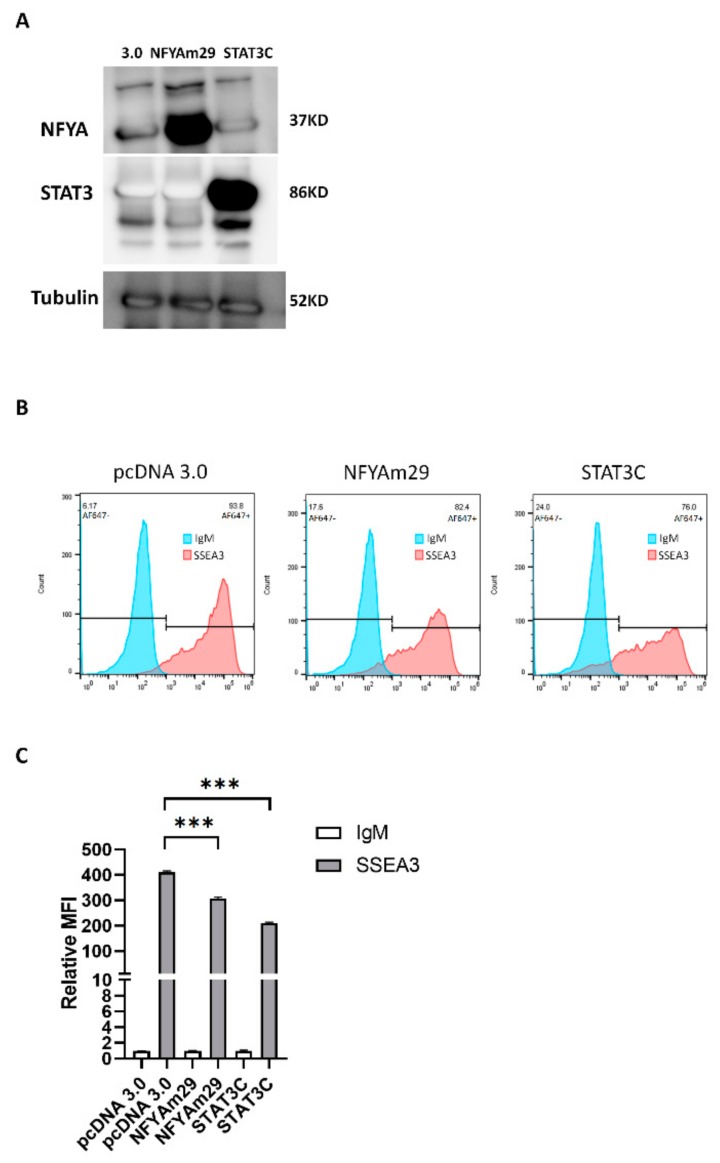
NFYAm29 and STAT3C repress SSEA3 synthesis in 2102Ep cells. (**A**) Overexpressed NFYAm29 and STAT3C were detected by western blotting. Tubulin served as the loading control. 3.0 means empty vector pcDNA 3.0. (**B**) Flow cytometry showing the amounts of SSEA3 on 2102Ep cells after transfection with pcDNA 3.0 or NFYAm29 or STAT3C. (**C**) Relative mean fluorescence intensity (MFI) after staining for SSEA3. The synthesis of SSEA3 was significantly reduced in cells transfected with pcDNA 3.0 containing the NFYAm29 or STAT3C sequence compared with cells transfected with an unmodified pcDNA3.0 (control) (***, *p* < 0.001).

**Figure 5 cells-09-00177-f005:**
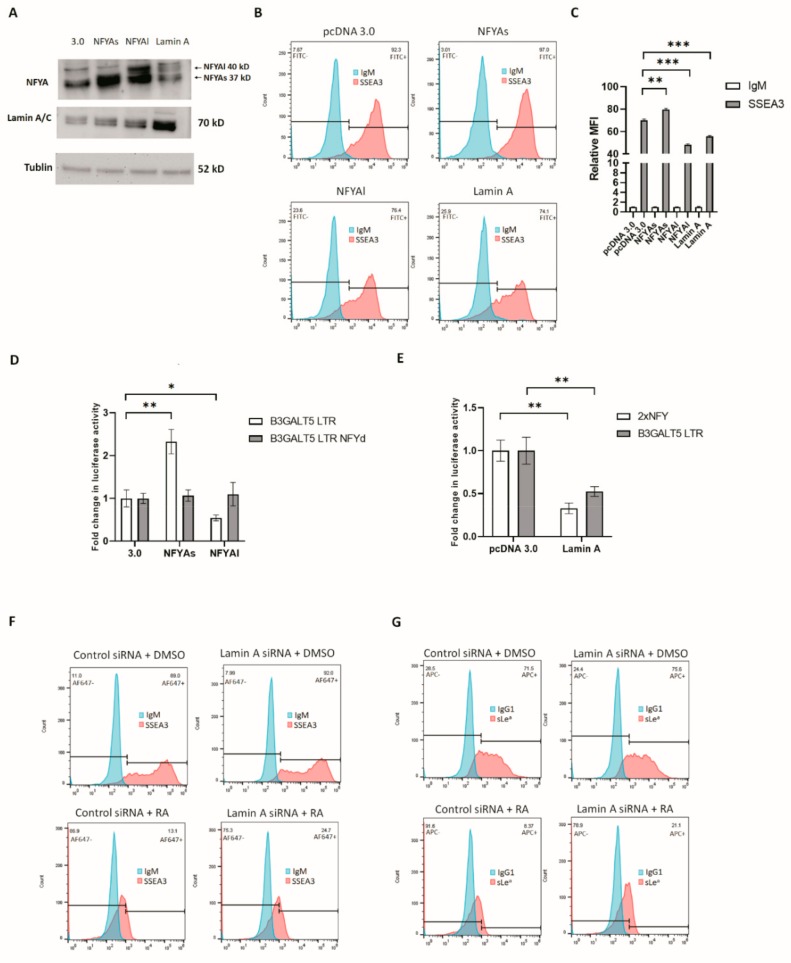
NFYAs, NFYAl, and lamin A regulate SSEA3 synthesis in 2102Ep cells. (**A**) NFYAs, NFYAl, and lamin A were detected by western blotting. Tubulin served as the loading control. (**B**) Flow cytometry showing the levels of SSEA3 on 2102Ep cells after transfection with pcDNA 3.0 or a pcDNA 3.0 containing an NFYAl, NFYAs, or lamin A sequence. (**C**) The MFI of stained SSEA3 compared with that of the controls. SSEA3 was enhanced in the presence of NFYAs and repressed in the presence of NFYAl and lamin A compared with the control (unmodified pcDNA 3.0). (**, *p* < 0.01; ***, *p* < 0.001) (**D**) NFYAs activates the *B3GALT5*-LTR promoter in 2102Ep cells, and NFYAl represses the activity of *B3GALT5*-LTR. Repression and activation were eliminated when the NFY-binding site was deleted in the *B3GALT5*-LTR promoter (*, *p* < 0.05; **, *p* < 0.01). (**E**) Lamin A represses the effects of the tandem repeat NFY consensus sequence and the *B3GALT5*-LTR promoter. (**, *p* < 0.01) (**F**) RA represses SSEA3 synthesis in NT2 cells, and this repression is partially reversed by lamin A siRNA. (**G**) RA represses sialyl Lewis a synthesis in NT2 cells, and this repression is partially reversed by lamin A siRNA.

**Figure 6 cells-09-00177-f006:**
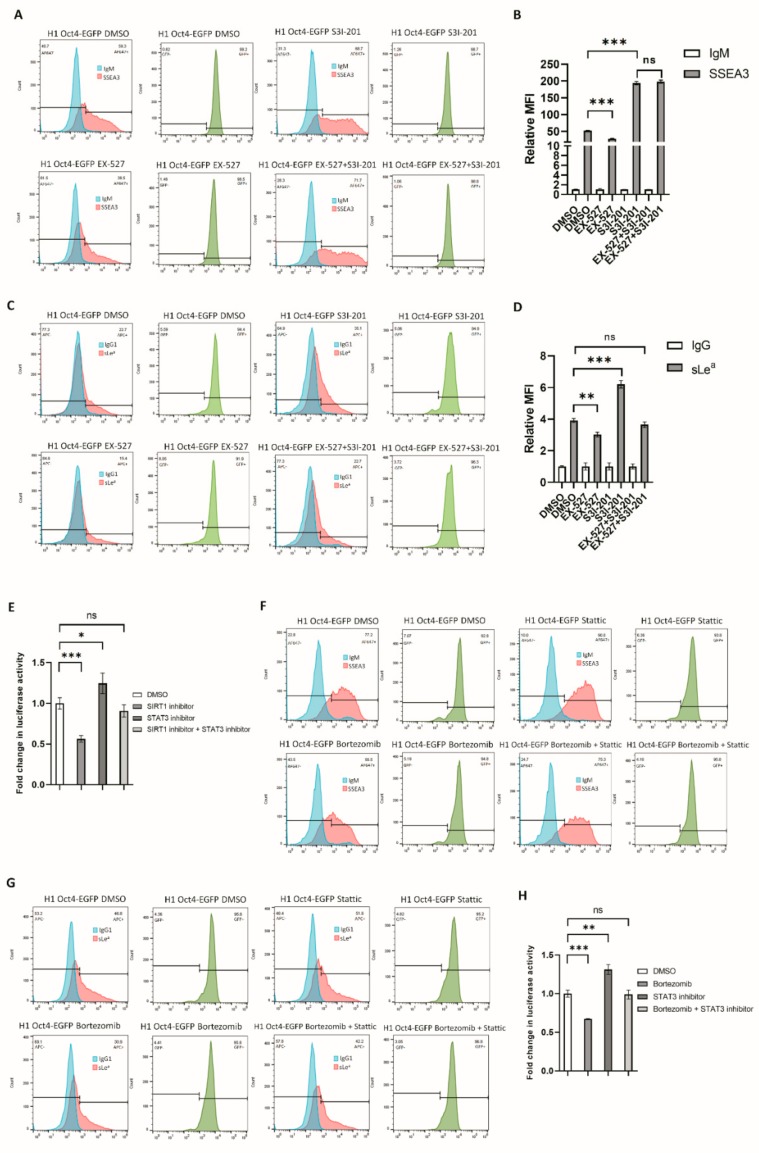
SSEA3 and sialyl Lewis a synthesis is altered by SIRT1 or STAT3 inhibitors. (**A**) Flow cytometry showing the levels of SSEA3 on H1 Oct4-EGFP cells after treatment with a SIRT1 or STAT3 inhibitor. All treated cells had similar GFP signal intensities under G418 selection. (**B**) The MFI of stained SSEA3 compared with the control. EX-527, a SIRT1 inhibitor, repressed SSEA3 synthesis in H1 Oct4-EGFP cells, and S3I-201, a STAT3 inhibitor, activated SSEA3 expression. EX-527-mediated repression of SSEA3 synthesis was reversed by S3I-201 (***, *p* < 0.001 and ns: not statistically significant). (**C**) Flow cytometry showing the levels of sialyl Lewis a on H1 Oct4-EGFP cells after treatment with a SIRT1 or STAT3 inhibitor. All treated cells had similar GFP signal intensities under G418 selection. (**D**) The MFI of stained sialyl Lewis a compared with the control. EX-527-mediated repression of sialyl Lewis a synthesis was reversed by S3I-201 (**, *p* < 0.01; ***, *p* < 0.001; and ns: not statistically significant). (**E**) EX-527 represses B3GALT5-LTR promoter activity in 2102Ep cells, and the EX-527-mediated repression of the B3GALT5-LTR promoter activity is reversed by S3I-201 (*, *p* < 0.05; ***, *p* < 0.001; and ns: not statistically significant). (**F**) Bortezomib represses SSEA3 synthesis in H1 Oct4-EGFP cells, and this repression is reversed by Stattic. (**G**) Bortezomib represses sialyl Lewis a synthesis in H1 Oct4-EGFP cells, and this repression is reversed by Stattic. (**H**) Bortezomib represses B3GALT5-LTR promoter activity in 2102Ep cells, and the Bortezomib-mediated repression of the B3GALT5-LTR promoter activity is reversed by Stattic (**, *p* < 0.01; ***, *p* < 0.001; and ns: not statistically significant).

**Figure 7 cells-09-00177-f007:**
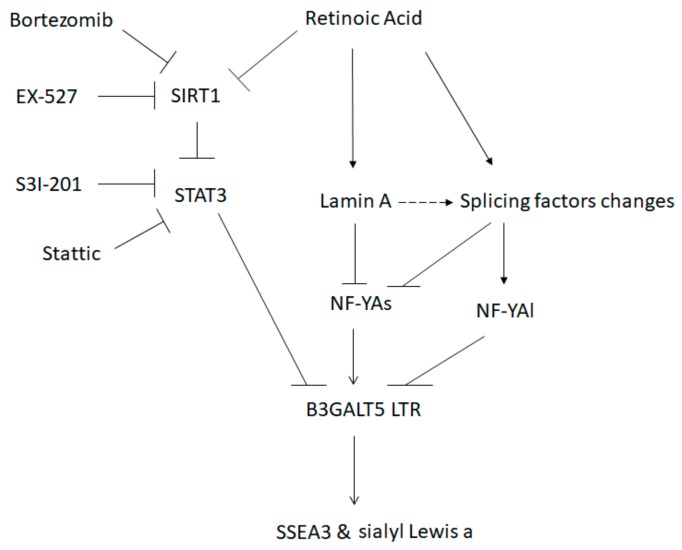
Signaling cascades of RA-mediated repression of *B3GALT5*-LTR promoter activity that reduce SSEA3 synthesis in human ES cells. RA induces two pathways that repress *B3GALT5*-LTR promoter activity, and thereby downregulates SSEA3 synthesis. One pathway involves RA-mediated SIRT1 repression and results in STAT3 signal enhancement, which causes the activated STAT3 to bind and repress the activity of the *B3GALT5*-LTR promoter, thereby reducing SSEA3 synthesis. The second pathway is RA-mediated enhancement of lamin A level, which changes the NFYAs/NFYAl ratio, thereby repressing *B3GALT5*-LTR promoter activity and reducing SSEA3 synthesis. Several changes in splicing factor levels (see Discussion) involve a lamin A-dependent or -independent pathway. The drugs used in this study are also shown in this schematic.

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
