# Peer review of "SSEA3 and Sialyl Lewis a Glycan Expression Is Controlled by B3GALT5 LTR through Lamin A-NFYA and SIRT1-STAT3 Signaling in Human ES Cells"

_cells, 2020, doi:10.3390/cells9010177_

Round 1
Reviewer 1 Report
The Authors appear to have responded to my previous comment on their use of discontinued antibodies.
Going through the text again, I notice that they use a plasmid called NFYAm29 (Fig. 4). They should give a reasonable explanation and they should describe it in the Methods section.
Author Response
Reviewer 1
The Authors appear to have responded to my previous comment on their use of discontinued antibodies.
Going through the text again, I notice that they use a plasmid called NFYAm29 (Fig. 4). They should give a reasonable explanation and they should describe it in the Methods section.
[Response]
We added explanation for NF-Y heterotrimer in Page 7 of the revised manuscript, and added description of NFYAm29, a dominant-negative mutant of NFYA, in Page 8.
Reviewer 2 Report
It is a well-done paper that could be accepted
Author Response
Reviewer 2
It is a well-done paper that could be accepted.
[Response]
Thank you very much.
This manuscript is a resubmission of an earlier submission. The following is a list of the peer review reports and author responses from that submission.
Round 1
Reviewer 1 Report
The manuscript of Cai et al attempts to dissect the transcriptional control of some key cell surface markers of embryonic stem cell renewal/differentiation such as SSEA3. Unfortunately the manuscript is not clearly written, lacks vital information and the presentation of the data is confusing so I found it extremely difficult to identify what is the overall question/focus of the study. The abstract and introduction provide little context related to the work and some of the statements are factually wrong and mis-referenced e.g. Lines 45-46: "SSEA3 is a marker related to the depletion of non-reprogramming cells, which are derived from induced pluripotent stem cells".
The authors overstate their findings e.g. they claim that B3GALT5-LTR is highly expressed in EC and ES cells at levels comparable to colon cancer cell lines but the data in fig 1A clearly show that this is not the case. Important information is also missing in the results sections (e.g. the origin of 210ep cells is not explained). The data quality and presentation in the figures requires dramatic improvement especially in terms of clarity and inclusion of appropriate controls. The number of biological repeats as well as the nature of assays used to obtain the data should be clearly stated. Finally there is a lack of consistency in terms of cell lines analysed (authors present data for different cell lines in each figure).
Reviewer 2 Report
The manuscript by Cai et al. describes the role of two transcription factors -NF-YA and STAT3- in the regulation of an LTR-driven transcript of B3GALT5 in ES cells, positive or negative depending from the differentiation status. It also add data on the role of LaminA and Sirt1 in this system.
The topic is interesting; the results are convincing and the conclusions are consequent to the findings. The manuscript is balanced and organized, generally well written. However, I do feel that there is one important aspect that should be dealt with.
The authors use for NF-YA and STAT3 studies (WBs and CHIPs) two Santa Cruz rabbit polyclonals -SC-10779 and SC-482- which have been discontinued. The present unavailability of such reagents prevents other scientists to reproduce -or continue studies from- the results shown in this manuscript. This is a general problem with many other SC antibodies, and not specific to this manuscript. However, as my understanding is that other antibodies are commercially available for these two transcription factors, I suggest repeating at least some of the analysis with other antibodies that can be actually purchased for future studies.